# "Restricted Geometry" Effect on Phase Transitions in KDP, ADP, and CDP Nanocrystals

**Vladislav V. Tarnavich [1], Alexander S. Sidorkin [2] , Tatiana N. Korotkova [3],**
**Ewa Rysiakiewicz-Pasek [4] , Leonid N. Korotkov [5],* and Nadezhda G. Popravko [2]**

[1] National Research Center Kurchatov Institute–Petersburg Nuclear Physics Institute, 188300 Gatchina, Russia; tarnavich@gmail.com

[2] Voronezh State University, Universitetskaya ploschad, 1, 394006 Voronezh, Russia; sidorkin@phys.vsu.ru (A.S.S.); nekalyaka@gmail.com (N.G.P.)

[3] Voronezh Institute of the RF Ministry of Internal Affairs, 394065 Voronezh, Russia; tn_korotkova@mail.ru

[4] Department of Experimental Physics, Faculty of Fundamental Problems of Technology, Wroclaw University of Science and Technology, 50-370 Wroclaw, Poland; ewa.rysiakiewicz-pasek@pwr.edu.pl

[5] Voronezh State Technical University, Moskovskii pr. 14, 394026 Voronezh, Russia

* Correspondence: l_korotkov@mail.ru

**Abstract:** The dielectric properties of composite materials prepared by the embedding of ferroelectrics potassium dihydrogen phosphate (KDP), cesium dihydrophosphate (CDP), as well as antiferroelectric ammonium dihydrogen phosphate (ADP) into porous glass matrices with an average size of through pores of 7, 46, and 320 nm have been studied. It was found that an increase occurred in the phase transitions temperature ($T_C$) for embedded particles in comparison with corresponding bulk materials. Some possible mechanisms of influence of "restricted geometry" on the Curie temperature are discussed. Estimates of $T_C$ shifting as a result of the "pressure effect" caused by elastic stresses in embedded particles as well as the result of bias electric field influence arising due to the piezoelectric effect are made. The possibility of using the tunneling Ising model to explain the experimental results is discussed.

**Keywords:** matrix composite; dielectric permittivity; Curie temperature; elastic stresses; tunneling effect; piezoelectric effect

## 1. Introduction

In recent years, the intensive development of nanotechnologies has provided a significant stimulus to clarify patterns of the size effect in systems with ultradisperse particles of different topologies and dimensions. As an example of such systems, consider composite materials that are polar dielectrics embedded in porous structures having a branched network of through pores. Here, the embedded substances form either a system of isolated particles or a complex dendritic structure determined by the size and topology of the pores in original matrix, as well as by the surface tension, wettability, etc. [1].

The effect of "restricted geometry" manifests differently in the case of various ferroelectrics. It is assumed [2] that a decrease in the sizes of the ferroelectric crystal leads to an increase of the surface energy part in the total energy of the particle, an increase in the energy of the depolarizing field, and the energy associated with mechanical and chemical interactions between the ferroelectric particles and the substrate or matrix.

A combination of these mechanisms can result in both a decrease and an increase in the temperature of the structural phase transition ($T_C$). For example, sodium nitrite embedded in a porous glass matrix with nanosize through pores demonstrates shifting of ferroelectric (FE) phase transition in the

low-temperature direction [3]. A similar situation occurs in the case of ammonium hydrogen sulfate, in which the temperatures of the both ferroelectric and antiferroelectric (AFE) phase transitions are decreased [4–7].

Alongside this, potassium dihydrogen phosphate [8–13], Rochelle salt [14,15], and triglycine sulfate [14,16–19], in contrast, are characterized by an increase of ferroelectric phase transition temperature under "restricted geometry" conditions. However the data on the magnitude of $T_C$ shifting as a function of the matrix mean that the pore diameter varied significantly for different researchers.

It was suggested [8] that the rise of the Curie temperature in KDP inclusions is the result of the tensile deformations which appear due to differences in the temperature coefficients of linear expansion (TKL) of the embedded substance and matrix material. This assumption was confirmed by a comparative analysis of shifting of the phase transition temperature in isomorphous crystals of the ferroelectric potassium dihydrogen phosphate and the antiferroelectric ammonium dihydrogen phosphate with the different baric coefficients ($\gamma = dT_C/dP$, where P is a hydrostatic pressure) and the different temperature coefficients of linear expansion [9].

However the expansion of the temperature range, within which the ferroelectric phase exists in the embedded nanoparticles of triglycine sulfate and Rochelle salt [14–19] cannot be related to the action of mechanical stresses. The phase transition temperatures of these salts do not differ so much from the temperature at which they were introduced into the matrix.

These circumstances indicate a strong "non mechanical" interaction between the matrix and the embedded material. However, mechanisms of such interaction have not yet been clarified.

This work is a comparative study of the dielectric properties of composite materials obtained by embedding the related H – bonded crystals potassium dihydrogen phosphate ($KH_2PO_4$ - KDP), cesium dihydrophosphate ($CsH_2PO_4$ - CDP), and ammonium dihydrogen phosphate ($NH_4H_2PO_4$ - ADP) in porous matrixes with different average pore diameters. The KDP, CDP, and ADP crystals have a similar chemical composition, but differ in the baric coefficients and in the temperature coefficients of linear expansion.

## 2. Samples Preparation and Validation

The KDP, CDP, and ADP polycrystalline samples were prepared by compacting appropriate powders in the form of disk with a diameter of 10 and a thickness of 1 mm. The grain sizes were within 0.1–0.4 mm.

Composite materials were fabricated by embedding corresponded salts into glass matrices with average diameter of through pores equal to approximately 320, 46, and 7 nm (the pore sizes were determined by the SEM method).

The samples of the glass matrices were cut off in the form of flat plates of about $10 \times 10 \times 0.7$ mm. The porosity of used glasses matrix, which ranged from 40% to 55%, was determined by the relative mass decrement method during the preparation of the samples. (The technology of porous glasses and methods of their certification are presented in references [7,20]).

Before filling, the glass matrices were annealed and weighed. Then they were immersed in a saturated aqueous solution with corresponding quantities of salt at a temperature of 90 °C for four hours.

After the filling, the samples were pulled out and dried at room temperature for 24 h. Thereafter, they were subjected to 6-h thermal treatment at a temperature of 150 °C. Then, the surface layer was mechanically removed. The share of the substance embedded in the matrix had around a 0.2 volume fraction.

The samples were labeled as follows: KDP-PGX, ADP-PGX, and CDP-PGX, where X was the average matrix pore diameter in nanometers.

The X-ray diffraction patterns for the PG-320 glass matrix, the original KDP powder, and the KDP-PG320 composite, obtained at room temperature using a x-ray diffractometer (Model-DRON-3) with Bragg–Brentano geometry, Cu K$\alpha_1$-radiation, are shown in Figure 1 a–c correspondingly.

One can see the gallo in the case of a glass matrix, which is characteristic for amorphous materials. Lines of intensity (I) in Figure 1b correspond to the reference data for potassium dihydrogen phosphate. The X-ray pattern obtained for the composite KDP-PG320 (Figure 1c) is a superposition of the two previous I(2θ) dependences. Thus, the crystal structure of the bulk and embedded material is identical, and there are no extraneous phases.

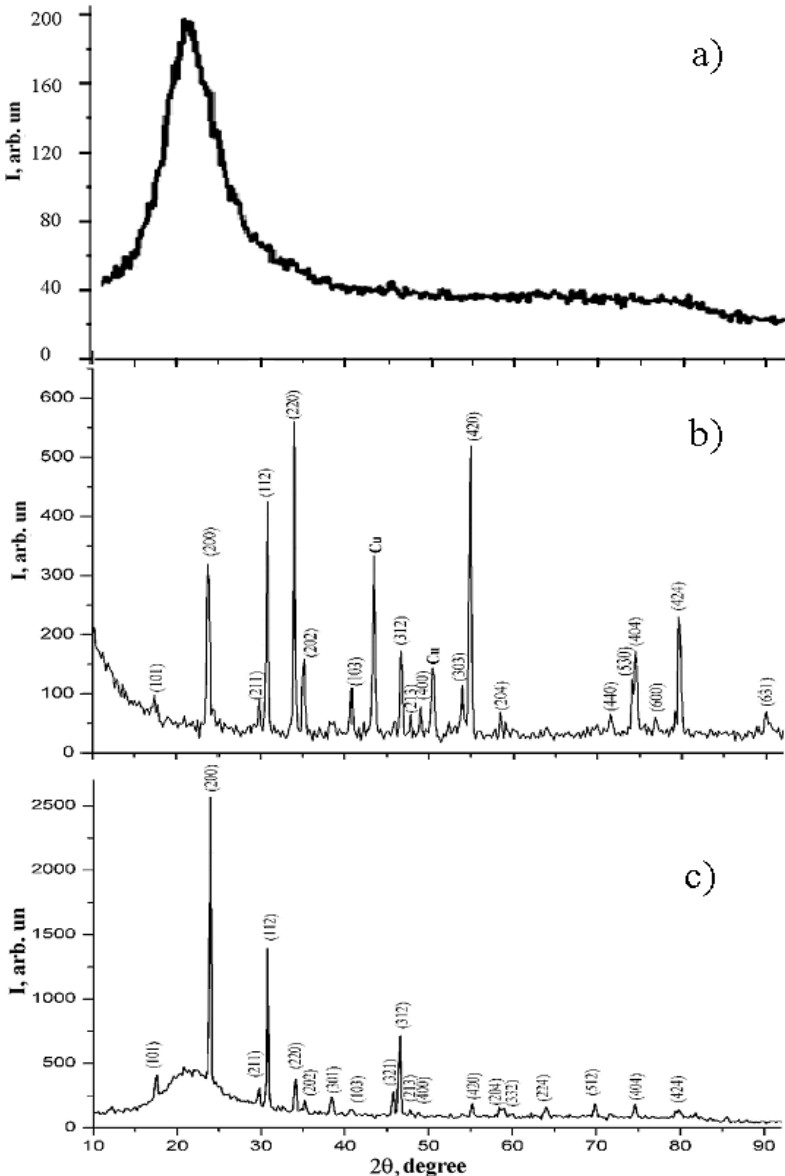

**Figure 1.** X-ray diffraction patterns for porous glass matrix PG320 (**a**), polycrystalline KDP (**b**) and the KDP-PG320 composite (**c**). The lines "Cu" on the panel "b" correspond to the copper cuvette.

For experiments, a conductive silver-containing paste was applied on large facets of the samples, as electrodes. The samples were placed in a cryostat, containing an evacuated measuring cell. Before each experiment, the samples were annealed about a one hour at a temperature of ≈ 100 °C to remove the adsorbed water. Its content in the sample was controlled qualitatively by measuring dielectric permittivity and the dielectric losses tangent (tgδ) during thermal treatment. The cessation of $\varepsilon'$ and tgδ variations with time indicated the removal of most of the water.

During the experiment, the temperature in the cryostat varied within 85–300 K, and its error measurement did not exceed ± 0.5 K. Measurements of the dielectric permittivity ($\varepsilon'$) were carried

out in a slow heating mode with a rate of about 0.5 K/min using the E7-20 immittance meter at the frequency f = 1 kHz.

## 3. Experimental Results

The temperature dependences of dielectric permittivity of ammonium dihydrogen phosphate, potassium dihydrogen phosphate, and cesium dihydrogen phosphate, as well as composites based on them, are shown in Figures 2–4. One can see that dielectric permittivity of the investigated polycrystalline samples ADP, KDP, and CDP is reduced in comparison with known references data [21]. This is due to significant porosity of the prepared samples, which, however, does not affect the corresponding Curie temperatures.

In the case of ammonium dihydrogen phosphate, an abrupt change of $\varepsilon'$ is observed near the $T_C \approx 150$ K due to the first-order AFE phase transition (Figure 2a).

For ADP-PG composites, as well as for bulk ADP, antiferroelectric phase transition was accompanied by step-like anomalies of the dielectric response (Figure 2b–d). These anomalies were visibly diffused as the average pore diameter of the matrixes is decreased. Apparently, this is a consequence of the phase transition diffuseness in embedded ADP particles.

It is known [21] that the appearance of an order parameter at FE or AFE phase transition leads to a decrease in dielectric permittivity. We used this to determinate $T_C$, which was found as the temperature below for which a decrease in $\varepsilon'$ was observed.

The temperature of AFE phase transition in ADP nano-particles, included in the composite, is slightly higher than $T_C$ of a bulk ammonium dihydrogen phosphate (Table 1).

The $\varepsilon'(T)$ dependencies for potassium dihydrogen phosphate and cesium dihydrogen phosphate polycrystalline samples, as well as composites based on them, are shown in Figures 3 and 4, respectively.

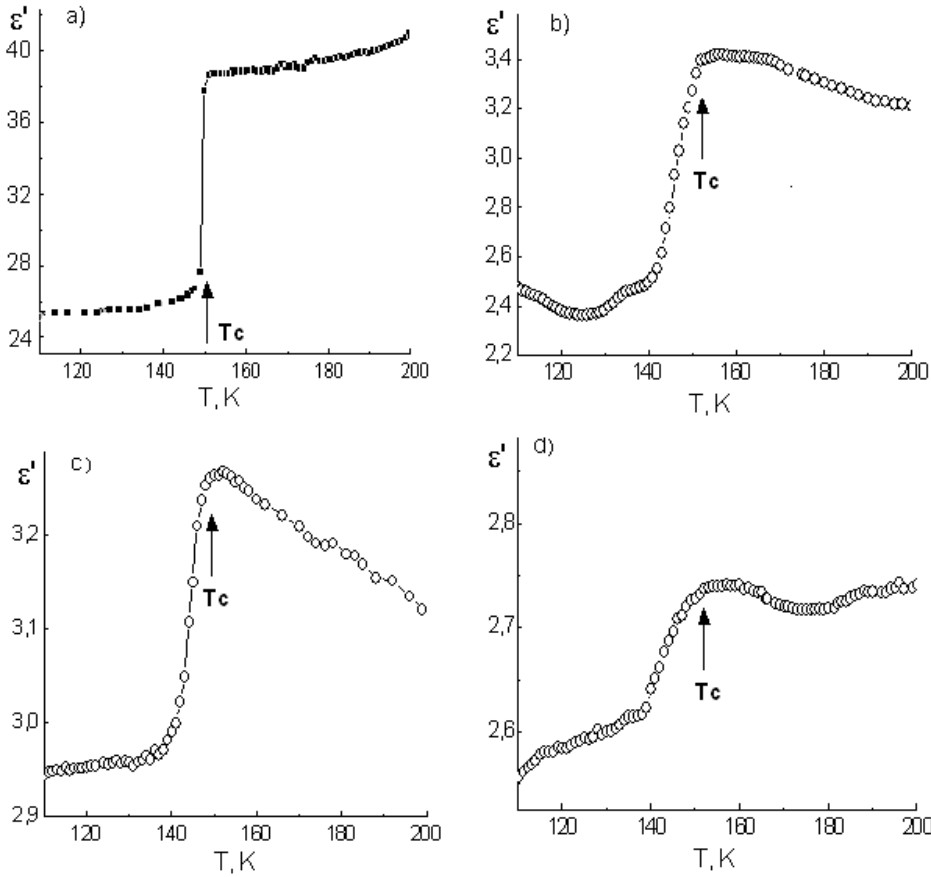

**Figure 2.** Temperature dependences of $\varepsilon'$ for polycrystalline ADP (**a**) and for ADP-PG320 (**b**), ADP-PG46 (**c**), and ADP-PG7 (**d**) composites, obtained at slow heating.

**Table 1.** The values of $T_C$ for the materials under study.

| Phase transition temperature | $T_C$, K (heating) | $T_C$, K (heating) | $T_C$, K (heating) | $T_C$, K (heating) |
|:---:|:---:|:---:|:---:|:---:|
| Pore sizes | Bulk | 320 nm | 46 nm | 7 nm |
| **ADP** | 150 | 152 | 149 | 152 |
| **KDP** | 122 | 125 | 122 | 130 |
| **CDP** | 154 | 157 | | 159 |

In the vicinity of the FE phase transition in the bulk KDP ($T_C \approx 122$ K) and CDP ($T_C \approx 154$ K) samples, distinct maxima of the dielectric permittivity were observed. Their position in the temperature axis is in good agreement with references [21,22].

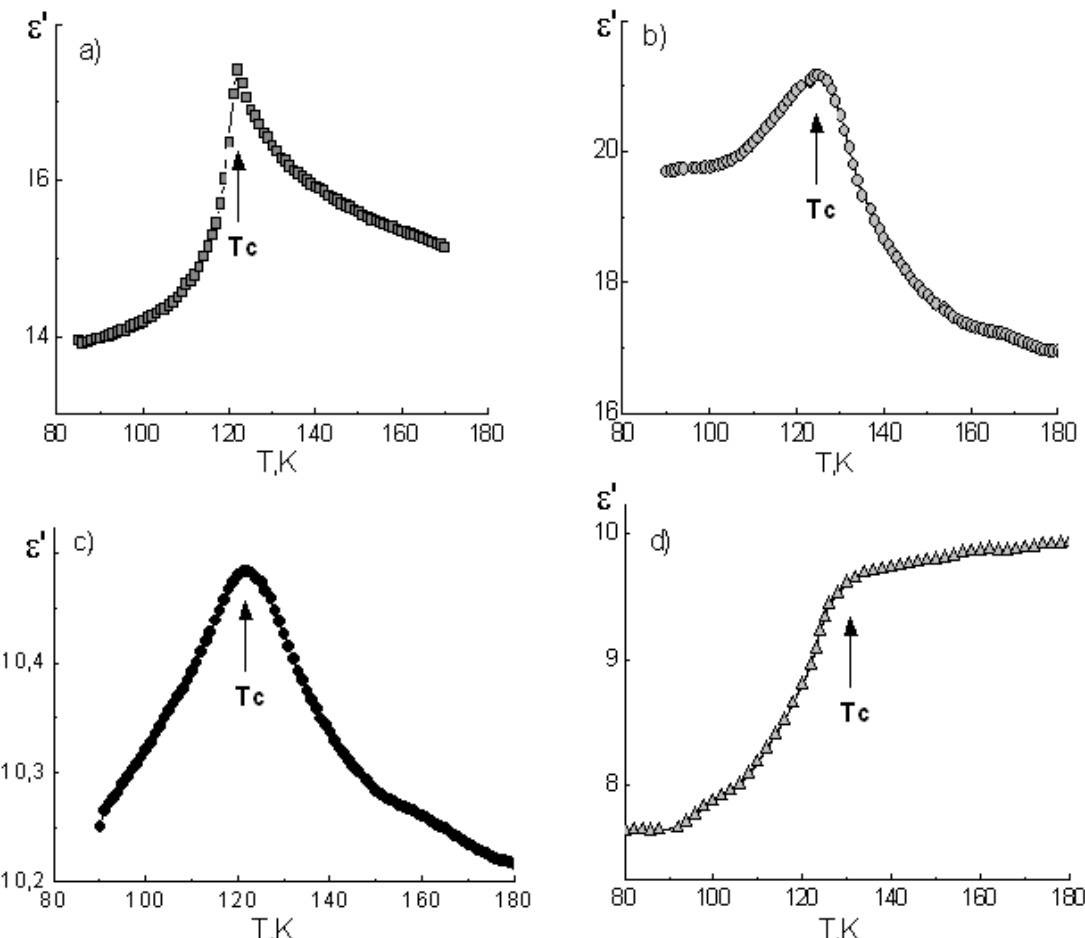

**Figure 3.** Temperature dependences of $\varepsilon'$ for polycrystalline KDP (**a**) and for KDP-PG320 (**b**), KDP-PG46 (**c**), and KDP-PG7 (**d**) composites, obtained at heating.

In the case of composite materials, $\varepsilon'$ maxima were smeared and shifted in the high-temperature direction with decreasing of the mean pore diameter of the glass matrices (Table 1). Obviously this is a result of increases in temperature and a smearing of the FE phase transitions in particles of embedded salts. The KDP-PG46 composite is the one exception here, because it has the same phase transition temperature as the bulk KDP.

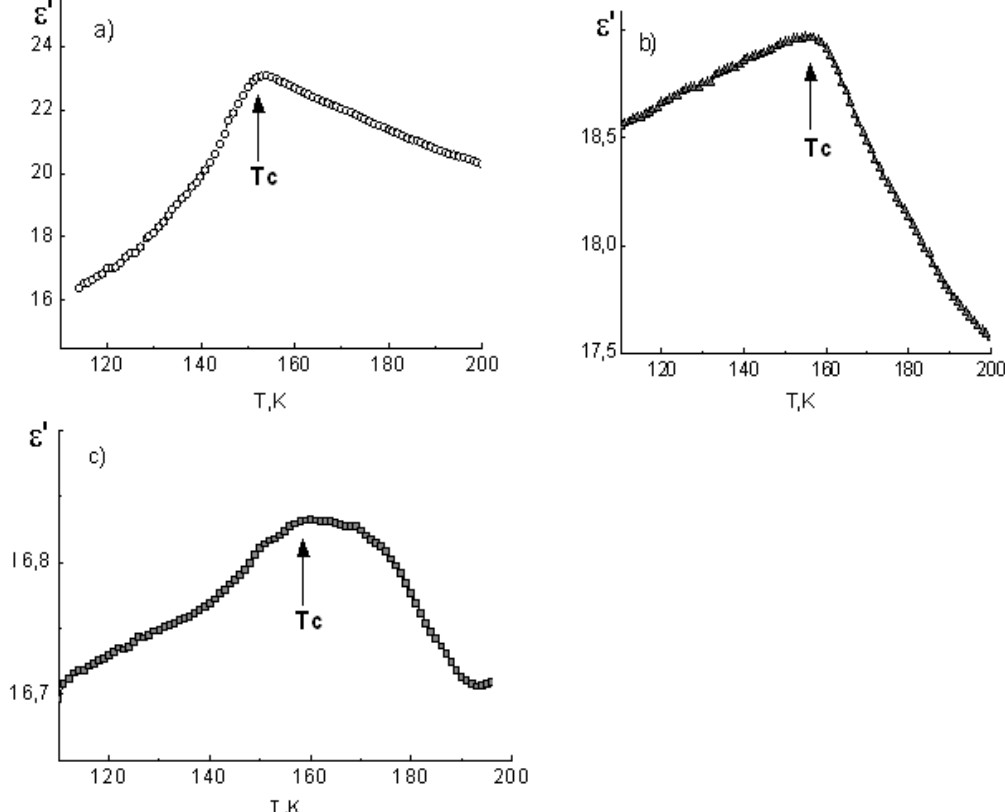

**Figure 4.** Temperature dependences of $\varepsilon'$ for polycrystalline CDP (**a**), CDP-PG320 (**b**), and CDP-PG7 (**c**) composites, obtained at heating.

Comparing the values of $T_C$ shifting for ADP, KDP, and CDP crystals under restricted geometry conditions, we can state that there was an increase in the phase transition temperature in embedded particles. The largest shifting of $T_C$ was found for the ferroelectric KDP, and the smallest one was found for the antiferroelectric ADP.

## 4. Discussion

An influence of the pore size on the phase transition temperature in KDP and ADP particles, embedded in porous matrices, has already been discussed [8–11,13]. However, experimental results obtained by different authors contain some contradictions. For example, the experimental study of the dielectric properties of KDP in porous structures with pore dimensions D = 7–100 nm demonstrate that a decrease in the pore diameter leads to a monotonous increase in the $T_C$. At the same time, the nonmonotonic behavior of the $T_C(D)$ dependence was revealed in references [10,11] on the basis of dielectric and dilatometric measurement results.

### 4.1. "Pressure Effect"

The results of our experiments showed an increase in the $T_C$ in ADP, CDP, and KDP particles embedded in porous glass matrices with D ≈ 320 nm (Table 1).

Following the authors of reference [8], we assumed that the observed increase in the phase transition temperatures is a consequence of mechanical tensile stresses in the embedded particles. These stresses are the result of differences between the temperature expansion coefficients (TKL) of the matrix and the embedded material, and arise during the cooling of the sample from the temperature, at which the filling matrices was carried out (≈ 370 K), to the $T_C$. We will call this phenomenon the "pressure effect".

The temperature coefficient of linear expansion for the quartz glass was estimated by the value $\alpha_{SiO2} < 10^{-6}$ K$^{-1}$ [23], whereas the coefficients $\alpha_i$ for ADP in the temperature range of 203–407 K were: $\alpha_1 = \alpha_2 \approx 34.0$–$39.3 \times 10^{-6}$ K$^{-1}$ and $\alpha_3 \approx 1.9$–$5.3 \times 10^{-6}$ K$^{-1}$ [24]. For KDP crystal in the temperature range of 123–363 K, the temperature coefficients were: $\alpha_1 = \alpha_2 \approx 20$–$26.6 \times 10^{-6}$ K$^{-1}$ and $\alpha_3 \approx 34.3$–$44.6 \times 10^{-6}$ K$^{-1}$ [24]. And for cesium dihydrogen phosphate, the same values are: $\alpha_1 \approx 31 \times 10^{-6}$, $\alpha_2 \approx 84 \times 10^{-6}$, and $\alpha_3 \approx -25 \times 10^{-6}$ K$^{-1}$ [25].

Silicon dioxide has a lower TKL value in comparison with ADP, CDP, and KDP, so the tensile stresses arise in embedded particles during cooling of the composite sample, and due to the compressive stresses in the matrix.

If pores of the matrix are partially filled, and the thickness of the "layer" of embedded substance is much smaller than the thickness of the interchannel space of the porous matrix, then the deformation of the interchannel space can be neglected. Under these conditions, the effective elastic stresses in particles of the embedded material may be estimated using the following expression:

$$\sigma_{ef} \approx \alpha_{av} c_{av} \Delta T \tag{1}$$

where $\alpha_{av} = (\alpha_1 + \alpha_2 + \alpha_3)/3$ is a mean TKL value, $c_{av} = (c_{11} + c_{22} + c_{33})/3$ is a mean value of elastic modulus of the embedded substance, and $\Delta T$ is the difference between the embedding temperature and the $T_C$.

We assume that the tensile stress $\sigma_{ef}$ affects on the phase transition in the same way as the hydrostatic pressure. Taking this into account, we can estimate the value of the phase transition shifting as:

$$\Delta T_C = T_C(\sigma_{ef}) - T_C(\sigma_{ef} = 0) \approx \sigma_{ef}\, \gamma \tag{2}$$

where $\gamma \approx -8.5 \times 10^{-8}$, $-4.5 \times 10^{-8}$, and $-3.4 \times 10^{-8}$ K/Pa for CDP, KDP, and ADP particles, respectively [21,22].

Using the reference data for ADP, KDP [24], and CDP [26,27] crystals, one can find the corresponding average values of the elastic modulus $c_{av} = 5.5 \times 10^{10}$, $7 \times 10^{10}$ and $4 \times 10^{10}$ Pa, the mean stress values $\sigma_{ef} \approx 2.06 \times 10^8$; $2.2 \times 10^8$ and $1.73 \times 10^8$ Pa, and to estimate the shifting values $\Delta T_C \approx 7$, 10, and 14.7 K for ADP, KDP, and CDP, respectively.

Thus, the obtained $\Delta T_C$ estimates are of the same order of magnitude as the values obtained from the experiment.

However, we should note that if thicknesses of embedded matter layers and the interporous space are comparable, then there will be noticeable deformations in the glass matrix, which will lead to a decrease in the elastic stresses in particles of the embedded material. This is most likely to occur for composites with a high content of embedded matter. The results of dilatometric measurements for composites ADP-PG160 and ADP-PG23 [11] have revealed a significant deformation of glass matrix in the AFE phase transition region, which illustrates this assumption.

Among the porous matrices used in these experiments, the PG46 matrices are markedly more brittle. This indicates thin interporous partitions in them. It is clear that that this circumstance led to a very small $T_C$ shifting in the KDP-PG46 composite in comparison with the bulk material (Table 1).

### 4.2. Transversal Tunneling Ising Model

A strong "pressure effect" in crystals of potassium dihydrogen phosphate family is related to the mechanism of the ferroelectric phase transition, initiated by the ordering of protons in two-minimal O-H . . . O hydrogen bonds [21]. Proton tunneling prevents their ordering and therefore significantly affects the temperature of the phase transition.

The dependence of $T_C$ on tunneling frequency $\Omega$, which is related to proton tunneling, can be described within the framework of Ising model in a transverse tunneling field, developed for a bulk KDP crystal [21,28]. This model can also be used to describe the phase transitions in CDP [29] and ADP [30].

According to references [21,28],

$$T_C = \Omega/\text{arcth}(\frac{2\Omega}{J_0}). \tag{3}$$

where $J_0$ is the energy of interaction of the i-th dipole, while with all other dipoles

$$J_0 = \sum_{R'} J(R,R') \tag{4}$$

where $J(R,R')$ is the energy of interaction of the dipole, located in the R-position, with another one, located in the R'-position.

Mechanical tensile stresses, acting on the embedded particles, lead to an elongation of the hydrogen bonds and, consequently, to a decrease in the tunneling frequency $\Omega$, that leads to an increase in the $T_C$.

Alongside this, the relation (3) makes it possible to explain the decrease in $T_C$ with a decrease in the size of embedded particles. For ultrafine particles, the sum (4) will decrease because of the limited number of interacting dipoles, which will lead to a decrease in $J_0$ and $T_C$.

### 4.3. Influence of the Piezoelectric Effect on the $T_C$

Let us discuss the possible impact of the piezoelectric effect on the $T_C$. Potassium dihydrogen phosphate and ammonium dihydrogen phosphate crystals are piezoelectrics in paraelectric phase with non-zero $g_{36}$ and $g_{14}$ piezoelectric coefficients, where $g_{36} >> g_{14}$ [24,31,32].

The piezoelectric effect can lead to the appearance of significant electric bias fields $E_3$ at the temperatures $T \geq T_C$ due to the appearance of the shear component of mechanical stresses $\sigma_6$ for inhomogeneous deformations of embedded particles.

We estimated the field strength $E_3 = g_{36}\sigma_6$ [31,32], and the possible increase in the $T_C$ as a result of its action [21]:

$$\Delta T^E_C(E) = T_C(E) - T_C(E=0) = AE_3 \tag{5}$$

Taking into account that coefficient $A \approx 5 \times 10^{-4}$ K·cm/kV for the KDP [33] and assuming that $\sigma_6 \approx 0.1\sigma_{ef}$, we can find $\Delta T^E_C \approx 13$ K.

Thus, the calculated shifting of the $T_C$ as a result of the electric field action for KDP, caused by the direct piezoelectric effect, is of the same order of magnitude as the experimentally observed value of $\Delta T_C$.

In the estimation of electric field $E_3$, we did not consider its decreasing due to the polarization of the glass matrix, as well as a charge leak owing to electrical conductivity of the material, although it is quite small at the temperatures near $T_C$ [34]. Therefore, it is obvious that the real values of the field $E_3$ can be less than the above estimates.

Phenomenological theory predicts the decreasing of antiferroelectric phase transition temperature under a bias electric field E [21]. However the experimental data regarding the electric field E effect on the $T_C$ in the ADP crystal are absent in references, probably due to the destruction of the bulk ADP samples below $T_C$ caused by strong elastic stresses [21]. At the same time, the experiments with antiferroelectric ADP - KDP solid solutions did not reveal affects of electric field on $T_C$, at least at E < 10 kV/cm [35].

The CDP crystal has a piezoelectric effect in the ferroelectric phase only [36]. Therefore, one can expect that the electric field acting in the CDP particles due to the piezoelectric effect will be significantly less than in KDP or ADP.

## 5. Conclusions

The experimental results have shown that the phase transition temperatures in ADP, KDP, and CDP particles embedded into porous glass matrices with a nanometer pore diameter (D) are higher than in

bulk samples. However, the observed relationship between Tc and D is ambiguous. In our opinion, this is due D not being the only factor affecting Tc.

An analysis of the factors that affect the $T_C$ in materials under study speaks in favor of two mechanisms providing the largest contributions.

The first one is caused by dependence of the $T_C$ on tensile strains in ferroelectric particles ("pressure effect") due to different TKL of the embedded material and the matrix. Its microscopic mechanism is the decreasing of proton tunneling due to the rise in the distances between the potential minima in double-well O-H . . . O hydrogen bonds.

The greatest increase in the $T_C$ due to the "pressure effect" should be expected for CDP particles, but the experimental value $\Delta T_C$ for CDP particles is smaller than the same for KDP, under equal experimental conditions.

Apparently, this is due to the bias electric fields induced by elastic stresses owing to the piezoelectric effect in the paraelectric phase of potassium dihydrogen phosphate. The CDP crystal has no piezoelectric effect above the $T_C$, so the phase transition temperature shifting in CDP particles is smaller than in the KDP.

The electric field induced by the piezoelectric effect in ADP particles, according to the above estimations, is almost double that in KDP. However it has no effect on the temperature of antiferroelectric phase transition.

Both of the above mechanisms are due to mechanical stresses, depending on both the difference between the TKL of embedded particles and the matrix, and the ratio of the layer thicknesses of the embedded substance and the interporous space. So, they depend on the filling of the matrix pores.

Together with the piezoelectric effect, another source of internal bias fields could be the chemical interactions of protons of the embedded salts and the oxygen atoms of the $SiO_2$ matrix. In particular, the existence of chemical interaction was revealed by analysis of IR spectra of nanocomposites based on particles of TGS and $SiO_2$ [37].

**Author Contributions:** Investigation, V.V.T., E.R.-P. and N.G.P.; Conceptualization, A.S.S.; Formal analysis, T.N.K.; Resources, E.R.-P.; writing—original draft preparation, L.N.K.

**Funding:** This work has been supported by the Russian Science Foundation (project N 17-72-20105).

**Conflicts of Interest:** The authors declare no conflict of interest.

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
