# Peer review of "“Restricted Geometry” Effect on Phase Transitions in KDP, ADP, and CDP Nanocrystals"

_crystals, doi:10.3390/cryst9110593_

Round 1

Reviewer 1 Report

see attached file

Reviewer 2 Report

The paper is devoted to the study of the size- and pressure- related effects on the phase transition temperature. The paper is organised well and will be interrsting to the readers of the Crystals journal. Authors demonstrate non-monotonic increase of the Tc with decrease of the pore size. The effect was discussed using three different theoretical approaches. However, few small remarks may be addressed by authors mainly aimed to improve the presentation:

1. Could you please add some brief description of the porous glass impregnation process. May be electronic microscopy of the samples is available.

2. Could you please organize all curves for each type of ferroelectric into one plot, not (a), (b), (c) and (d) if possible otherwise please add some vertical lines as a guidelines.

3. The absolute values of the dielectric permittivity should be also disussed. In particular, the sample KDP - PG320 demonstrate higher values of epsilon, that the polycrystal. While other samples show decrease.
